# Central Disorders of Hypersomnolence: Association with Fatigue, Depression and Sleep Inertia Prevailing in Women

**DOI:** 10.3390/brainsci12111491

**Published:** 2022-11-03

**Authors:** Sona Nevsimalova, Jelena Skibova, Karolina Galuskova, Iva Prihodova, Simona Dostalova, Eszter Maurovich-Horvat, Karel Šonka

**Affiliations:** 1Department of Neurology, Center of Clinical Neurosciences, General University Hospital, Charles University, Katerinska 30, 128 00 Prague, Czech Republic; 2Unit of Statistics, Institute of Clinical and Experimental Medicine, Videnska 1958, 140 21 Prague, Czech Republic

**Keywords:** narcolepsy type 1 and 2, idiopathic hypersomnia, excessive daytime sleepiness, disease severity, fatigue, depression, sleep inertia, sex differences

## Abstract

Fatigue, depression, and sleep inertia are frequently underdiagnosed manifestations in narcolepsy and idiopathic hypersomnia. Our cross-sectional study design included diagnostic interview accompanied by assessment instruments and aimed to explore how these factors influence disease severity as well as to elucidate any sex predisposition. One hundred and forty-eight subjects (female 63%) were divided into narcolepsy type 1 (NT1; n = 87, female = 61%), narcolepsy type 2 (NT2; n = 22, female = 59%), and idiopathic hypersomnia (IH; n = 39, female = 69%). All subjects completed a set of questionnaires: Epworth Sleepiness Scale (ESS), Hospital Anxiety and Depression Scales (HADS), Fatigue Severity Scale (FSS), and Sleep Inertia Questionnaire (SIQ). In narcoleptic subjects, questionnaire data were correlated with the Narcolepsy Severity Scale (NSS), and in subjects with idiopathic hypersomnia, with the Idiopathic Hypersomnia Severity Scale (IHSS). The highest correlation in narcoleptic subjects was found between NSS and ESS (r = 0.658; *p* < 0.0001), as well as FSS (r = 0.506; *p* < 0.0001), while in subjects with idiopathic hypersomnia, the most prominent positive correlations were found between IHSS and SIQ (r = 0.894; *p* < 0.0001), FSS (r = 0.812; *p* < 0.0001), HADS depression scale (r = 0.649; *p* = 0.0005), and HADS anxiety scale (r = 0.528; *p* < 0.0001). ESS showed an analogic correlation with disease severity (r = 0.606; *p* < 0.0001). HADS anxiety and depression scores were higher in females (*p* < 0.05 and *p* < 0.01), with similar results for FSS and SIQ scales (*p* < 0.05 for both), and a trend toward higher ESS values in females (*p* = 0.057). Our study illustrates that more attention should be focused on pathophysiological mechanisms and associations of fatigue, depression, as well as sleep inertia in these diseases; they influence the course of both illnesses, particularly in women.

## 1. Introduction

Central disorders of hypersomnolence are a group of disorders in which the primary complaint is excessive daytime sleepiness (EDS) not caused by disturbed nocturnal sleep, sleep apnea, insufficient sleep syndrome, and/or misaligned circadian rhythms. A low level of alertness and particularly EDS leads to reduced efficiency, diminished concentration, and poor memory during daytime activities and have an important impact on the quality-of-life of affected patients. Personal, educational, and occupational difficulties are frequent complaints, and patients may be predisposed to various somatic and psychiatric comorbidities [1,2]. 

The primary central disorders of hypersomnolence include narcolepsy, idiopathic hypersomnia, and Kleine–Levin syndrome. The international classification of sleep disorders [1] also includes hypersomnia due to medical and psychiatric disorders, hypersomnia associated with medications or substances, and insufficient sleep syndrome. 

Narcolepsy is the most common disease of the primary central hypersomnolence disorders and is characterized by a variety of manifestations such as imperative sleepiness, cataplexy, hypnagogic hallucinations, sleep paralysis, and disturbed nocturnal sleep. According to current classification [1], narcolepsy is divided into type 1 (narcolepsy with cataplexy) and type 2 (narcolepsy without cataplexy). Abnormal expression of rapid eye movement (REM) sleep during nocturnal and particularly daytime sleep presenting during the multiple sleep latency test (MSLT) characterizes both form of the disease. Narcolepsy type 1 (NT1), the more frequent form is a homogeneous entity with hypocretin deficiency while narcolepsy type 2 (NT2) is a heterogeneous disorder with an absence of clear biomarker predisposing to a challenging diagnosis [3]. Idiopathic hypersomnia (IH) is less frequent and is characterized by prolonged sleep attacks during the day, frequently accompanied by prolonged nocturnal sleep and difficulties awakening, termed sleep inertia [4]. Kleine–Levin syndrome (KLS) is an orphan disorder characterized by severe episodic hypersomnia with cognitive impairment accompanied by apathy or disinhibition [5]. In all three diseases, severe sleepiness may cause great handicap in the personal and social life of affected patients, and frequently may predispose to psychiatric comorbidities [5,6].

Daytime sleepiness, and particularly EDS, has also been studied regarding sex and gender prevalence. Results are not uniform; however, most studies have found a higher prevalence of EDS in women [7,8]. The prevalence of EDS, as measured by the Epworth Sleepiness Scale (ESS), has been found to be higher in the population of psychiatric patients [9], and again prevailing among female psychiatric patients. Sleep inertia is defined as the transitional state between sleep and wakefulness accompanied by compromised cognitive and physical performance, reduced vigilance, and the desire to return to sleep, and occurs also more frequently in women in the healthy population again [10].

The influence of sex on central disorders of hypersomnolence remains poorly understood. IH with long sleep duration is the only clinical entity from this group of disorders with a clear female predominance [11,12]. Historically, male predominance in KLS was assumed; however, Ambati et al. [5], in a cohort of 673 cases collected over 14 years, showed that the *TRANK1* gene region may predispose to difficult birth and to KLS without any relation to neonatal sex. 

Although men and women share an equal risk of developing NT1 [13], sex differences in manifestations have been reported. Studies have suggested that gender differences emerge around the age at initial manifestation, in which cataplexy appears at an earlier age in women [14,15]. Narcoleptic women have also been shown to be more objectively sleepy as measured by MSLT [16]. These results correlate with a recently published animal study in a narcolepsy mouse model [14]. Delta attacks in NT1 mice, likely analogous to sleep attacks in patients with NT1, as well as frequency of cataplectic attacks were more frequent in female mice. Recently, Gool et al. [17] published a cluster analysis of 1078 unmedicated adolescents and adults with central disorders of hypersomnolence from the EU-NN database and found a striking female preponderance within cluster 4 in combination with mild cataplexy, hypnagogic hallucinations, and sleep paralysis. The authors hypothesized as to the existence of a female NT1 subtype. 

Our center for sleep and vigilance disorders has focused on central disorders of hypersomnolence for many years. Narcolepsy, idiopathic hypersomnia, and Kleine–Levin syndrome are historically considered as neurological diseases in our country, while hypersomnia due to medical and psychiatric disorders, hypersomnia associated with medication or substances, and insufficient sleep syndrome need psychiatric care. The authors of the present article are neurologists, and therefore, our attention is focused on these three main hypersomnolence groups. We have a great deal of interest in any potential sex-related differences with respect to clinical course and severity of illness in our patients, as well as any related factors that may be relevant and more frequent psychiatric comorbidities. 

The primary aim of our study was to examine a group of patients with narcolepsy and idiopathic hypersomnia who visited our center for sleep and vigilance disorders within the last 2 years on an out-patients basis or as newly diagnosed cases and to find if fatigue, depression, and sleep inertia influence the course of both illnesses. No self-selection of the participants by authors was done. Secondary objectives were focused on gender differences. The patients simultaneously participated in a study focused on metabolon (the Health Research Agency, Grant Number NU20-04-00088), the results of which will be published elsewhere in the future. The present study is focused only on clinical features without any connection with their metabolome structure. 

## 2. Material and Methods

### 2.1. Participants and Study Design

Subjects who fulfilled the NT1, NT2, or IH diagnostic criteria according to ICSD-2 [18] or ICDS-3 [1] and were diagnosed at our department were invited to undergo a clinical examination by a neurologist specialized in sleep medicine and to complete a set of questionnaires. At the time of initial diagnosis, the diagnostic procedure included a detailed clinical examination, nocturnal video-polysomnography (PSG), preceded by 2-weeks actigraphy (and/or sleep diary), and followed by MSLT. Hypocretin examination was offered voluntarily and primarily to patients suffering from narcolepsy. Most subjects with IH continued after MSLT by sleep “ad libitum” (22 h), and in the remainder, actigraphic monitoring (and/or sleep diary) defined the sleep duration within a 24 h period. IH patients with long night-time sleep were characterized by sleep duration exceeding 10 h per night accompanied by long daily naps [18] or sleep duration at least 11 h or more within the 24 h period [1].

One hundred and forty-eight patients participated in the present study. The NT1 group consisted of 87 patients (mean age 33 ± 12.7 years, female 63% (n = 53)), NT2 included 22 patients (mean age 35.5 ± 14.3 years, female 59% (n = 13)), and IH was diagnosed in 39 patients (mean age 41 ± 13.2, female 69% (n = 27)). Females were predominant particularly among patients with long sleep duration. 

All subjects provided signed informed consent before participation, and the study was approved by the General Hospital ethics committee (No 15/19, Grant AZV VES 2020 VFN).

### 2.2. Diagnostic Interview Content and Assessment Instruments

Sleep specialists (KS, SN, IP, EMH, SD) led the interview, information about disease history was confirmed with patient documentation and, if necessary, completed. Neurologic examination was complemented body mass index (BMI) determination in each participant. All patients were asked to provide information about family history of the disease and comorbidities, particularly of psychiatric origin (anxiety, depression).

The participants completed a set of questionnaires and scales scoring sleepiness including the Epworth Sleepiness Scale as the subjective score of excessive daytime sleepiness (ESS) [19], NT1 and NT2 patients were asked to complete the Narcolepsy Severity Score (NSS) [20], and patients with IH the Idiopathic Hypersomnia Severity Score (IHSS) [21]. All patients completed the Hospital Anxiety and Depression Scale (HADS) [22], Fatigue Severity Scale (FSS) [23], and Sleep Inertia Questionnaire (SIQ) [24].

### 2.3. Statistical Analyses

Data were expressed as mean ± standard deviation for continuous variables and as number and percentage for categorical variables. A chi-square test was used for categorical variables. If chi-square test assumptions were invalid, Fisher’s exact test was performed. Student’s t-test was conducted for normally distributed continuous variables to compare the two groups. For non-normally distributed continuous data, the Mann–Whitney rank sum test was used. Differences between three diagnoses (NT1 × NT2 × IH) were assessed using One-way Analysis of Variance (ANOVA) and the Newman–Keuls test for pairwise comparisons. Interactions of comorbidities were analyzed using Two-way Analysis of Variance (e.g., classification according to sex and diagnosis). Relations between measured variables were expressed by correlation coefficient. Statistical significance was set at *p* < 0.05. Microsoft Excel and MedCalc software Version 20.116. (MedCalc Software Ltd., Belgium) was used for statistical analyses. 

## 3. Results

### 3.1. Demographic Data and Questionnaire Results

The demographic data and questionnaire results in all patients divided according to their clinical diagnosis is illustrated in Table 1. The mean age of participants was 35.4 ± 13.4 years, NT1 and NT2 patients were younger than IH subjects. Women prevailed slightly in all diagnostic categories. Body mass index (BMI) was the highest in the NT1 group (30.3 ± 7.6; *p* < 0.01); NT2 and IH patients did not differ in BMI.

Regarding questionnaire data, the HADS anxiety scale did not differ between the examined groups, while HADS depression scale was higher in IH versus NT1 patients (*p* < 0.05); no difference was found between NT2 and IH groups. Subjective sleepiness measured by ESS was significantly higher in NT1 versus NT2 (*p* < 0.01) and IH patients (*p* < 0.01); a difference between NT2 and IH subjects was not detected. The severity of symptoms (NSS) was greater in NT1 than NT2 patients (*p* < 0.001). Fatigue scale (FSS) did not differ on follow; the sleep inertia score (SIQ) was higher in IH versus NT1 patients (*p* < 0.01), while no difference was found between NT2 and IH groups. 

Table 2 illustrates differences between IH patients without (IH1) or with (IH2) long sleep duration per 24 h. IH2 patients had more severe disease symptoms (*p* < 0.001). While both groups did not differ in HADS anxiety or depression scales, IH2 patients had higher fatigue (*p* < 0.05), increased sleepiness measured by ESS (*p* < 0.05), as well sleep inertia (*p* < 0.05). 

The next step was to analyze correlation in disease severity (NSS and IHSS) and questionnaire data. We found the highest correlation between NSS and ESS (r = 0.658; *p* < 0.0001), and FSS (r = 0.506; *p* < 0.0001) (Figure 1). NSS also correlated less positively with SIQ (r = 0.325; *p* = 0.005), HADS anxiety scale (r = 0.217; *p* = 0.024), and HADS depression scale (r = 0.265; *p* = 0.005). The most prominent positive IHSS correlations were found with SIQ (r = 0.894; *p* < 0.0001), HADS depression scale (r = 0.649; *p* = 0.0005) and HADS anxiety scale (r = 0.528; *p* < 0.0001). FSS (r = 0.812; *p* < 0.0001 and ESS (r = 0.606; *p* < 0.0001) also showed an analogic effect on the disease severity (Figure 2).

In pooled data (NT1, NT2, and IH), we also found significant relationships between different questionnaire scales. SIQ values correlated positively with HADS anxiety (r = 0.523; *p* < 0.0001) and HADS depression (r = 0.595; *p* < 0.0001) scales. No correlation was found between ESS and HADS anxiety (r = 0.075); however, a slight positive correlation was noted between ESS and HADS depression (r = 0.192; *p* < 0.05). Correlation was also found between ESS and FSS (r = 0.360; *p* < 0.0001) and ESS and SIQ (r = 0.309; *p* < 0.0001). 

Regarding genetic predisposition, a positive family history for sleepiness was reported by six patients (6.8%) from the narcolepsy group; however, a diagnosis of narcolepsy was verified in only one case (son NT1 and mother NT2). Ten out of thirty-nine IH patients (25.6%) reported a positive family history of EDS. No apparent differences were found between cases with or without a positive family history. 

### 3.2. Sex Differences in the Revealed Data and Assessment Instruments 

In pooled data, we did not find any significant differences in disease severity; however, HADS anxiety and depression scales were higher in women in comparison to men (*p* < 0.05, resp. *p* < 0.01), similar results were found in FSS and SIQ scales (*p* < 0.05 for both). A trend toward female predominance was further shown in ESS (*p* = 0.057).

Questionnaire data analyzed according to clinical diagnosis is shown in Table 3. The results surprisingly showed differences only in the NT2 group, where women had higher scores in HADS anxiety (*p* < 0.05), fatigue (*p* < 0.01), sleepiness measured by ESS (*p* < 0.01) as well as sleep inertia (*p* < 0.05). No differences were found in NT1 and IH groups. Sex-related differences between IH1 (9 men, 12 women) and IH2 groups were very difficult to determinate due to low number of men in the IH2 group (3 men versus 15 women). Only FSS was higher in women (*p* < 0.05).

### 3.3. Comorbid Psychiatric Diseases

Considering anxiety and depression results from questionnaire data, we questioned the prevalence of anxiety, depression, or mixed anxiety depressive disorder, as well as other psychiatric disease requiring psychiatric care, in the history of our patients. Psychiatric history results are shown in Table 4. More than 40% of patients had a history of psychiatric care, more frequently IH patients in comparison to NT1 and NT2 patients. A history of mixed anxiety-depressive disorder was present in 41% of IH patients, in comparison to 22% of NT1 patients (*p* < 0.05). Regarding sex-related differences, in pooled data, mixed anxiety-depressive disorder was found more frequently in women than men (36.6% versus 16.4 %; *p* < 0.01). Behavioral developmental disorder (autism spectrum disorder or attention deficit hyperactivity disorder) was diagnosed in six patients with narcolepsy, and schizophrenia occurred in two NT1 patients during their follow-up period.

## 4. Discussion

The leading manifestation of central disorders of hypersomnolence is increased daytime sleepiness leading to impairment of vigilance, attention, concentration, performance, as well as short- and long-term memory. EDS, therefore, adversely affects productivity at work and school, quality of life, and social interactions, and may even increase morbidity and mortality [25]. Most patients with primary sleep disorders also experience fatigue. According to Veauthier et al. [26] younger age, female gender, and a high number of awakening and arousals have been shown to be predictive for fatigue. In narcolepsy, the number of patients suffering from fatigue may reach up to 63% [27]. Mood symptoms, particularly depression, have also been frequently reported in hypersomnia disorders of central origin, and their relation is often bidirectional [28]. Depressive mood and sleep inertia are closely connected as well [24]. Fatigue, depression, and sleepiness overlap not only in sleep disorders but also in other medical illnesses [29]. 

Our study aimed to elucidate how and by what extent the above-mentioned factors influence the course of central disorders of hypersomnolence, if there are any related differences between patients with narcolepsy and idiopathic hypersomnia, and if there are any sex-related differences in our patients.

The slight predominance of females in our cohort may be explained by their concurrent participation in the metabolon research project, in which men more frequently declined to participate. More frequent NT1 than NT2 or IH diagnosis correspond to known epidemiological prevalence [1]. 

Our study showed the highest BMI and excessive daytime sleepiness in the NT1 group, while the difference between NT2 and IH groups was not significant. Cataplexy, a typical sign of NT1, comprehensibly influenced the disease severity as measured by NSS. Conversely, the highest depressive mood and sleep inertia were found in IH, while no differences were detected between IH and NT2 patients. Dividing IH patients into an IH1 group without and IH2 group with long sleep duration/24 h showed more similarities between NT2 and IH1 patients. Particularly, ESS and SIQ values in patients without long sleep duration were like values in NT2 patients. Both groups are suggested to be heterogenous [4,30], and from time to time, the diagnosis should be retested and eventually changed from one to the other. IH patients with long sleep duration represent a more severe form of the disease and more stable diagnosis.

Interesting results were found between measured variables in the pooled data of patients with central disorders of hypersomnolence; the greatest correlation was found between sleep inertia with depression. Additionally, and reported previously [24], we found similar correlation with anxiety. Correlations between sleep inertia with increased daytime sleepiness, and daytime sleepiness with fatigue were less prominent. 

Correlation between severity scales in both diseases differed. In narcolepsy, the highest correlation with NSS was found with increased sleepiness and fatigue. Comparable results were also recently reported by Buskova et al. [31]. Idiopathic hypersomnia and its IHSS showed the highest correlation with sleep inertia followed by fatigue, depression, daytime sleepiness, and anxiety. A close connection between sleep inertia and depression has been published [24], and our findings confirmed this association too. These results may support antidepressants administration in patients suffering from idiopathic hypersomnia and provoke an interest in effective sleep inertia treatment.

Regarding sex-related differences, no clear differences with respect to diseases’ severity were found in pooled data; however, women had a greater deal of all measured variables including depression, anxiety, fatigue, sleep inertia, and a trend in daytime sleepiness. Interestingly, the most impressive predominance of these variables was found in women with NT2, in which a fragile background of the diagnosis has been suggested by some authors [30]. A significant prevalence of women in IH patients with long sleep duration, also verified by other authors [11,12], did not allow for the comparison of sex-related differences in this group. 

Psychiatric comorbidities are frequently studied in association with central disorders of hypersomnia, particularly with narcolepsy. According to Gudka et al. [32], mood disorders and anxiety are the most frequently reported comorbidities in narcolepsy; they even appear more frequently than obesity and metabolic disorders. It has been suggested that psychiatric manifestations are either a result of the chronic disabling nature of the disease or may represent a shared pathophysiology or a combination of both [6,33]. Hypocretin/orexin dysfunction in NT1 patients shows an increasingly important role in neuropsychiatric manifestations, particularly in depression and anxiety [34]. Hypocretin/orexin deficit is also one plausible etiological factor in the comorbidity of NT1 with schizophrenia [35,36]. 

Less attention has been given to the combination of psychiatric comorbidities with idiopathic hypersomnia. According to Arnulf et al. [37], diseases of the central nervous system, mostly mood disorders, may be up to twelve times more prevalent in IH patients in comparison with the general population. However, the differentiation of idiopathic hypersomnia depression from hypersomnia associated with psychiatric disorder may sometimes be difficult and require skilled clinical sleep examination [38]. 

Our results also confirmed frequent psychiatric comorbidities in patients with central disorders of hypersomnolence. A history of psychiatric care and treatment was reported by more than 40% of our patients. Mixed anxiety-depressive disorder was the most frequent diagnosis, more often affecting women and particularly IH patients. Comorbidity with neurodevelopmental diseases (attention deficit/hyperactivity disorder and pervasive developmental disorder) similarly to cases of schizophrenia were found only in the group of narcoleptic patients. 

The limitation of our study consists in the rather small cohort of our patients due to the rare occurrence of diseases despite the fact that our center includes a great majority of patients with central disorders of hypersomnolence from the Czech Republic. Another limitation is due to heterogeneous medication used in our patients. Only 40 subjects (27%) were free of medication. The majority of NT1 patients were on modafinil and antidepressants, thirteen of them used sodium oxybate, and one pitolisant. NT2 patients were medicated mostly by modafinil and some of them by methylphenidate. Modafinil or methylphenidate were used in IH patients too; however, their effect was less effective. The medication in all subjects was stable for months up to years. While narcolepsy is supposed to be predominantly “a disorder of falling asleep”, in idiopathic hypersomnia joins frequently “a disorder of awakening”. The therapeutic attention should be focused particularly on the improvement of sleep inertia in the IH group of patients.

## 5. Conclusions

The present study shows the important role of fatigue, depression, and sleep inertia in the clinical picture of central disorders of hypersomnolence in addition to their primary manifestation, i.e., excessive daytime sleepiness. Depression is highly correlated with sleep inertia which is the leading manifestation of patients with idiopathic hypersomnia; however, our questionnaire data showed that sleep inertia also affects patients suffering from narcolepsy. The pathophysiological connection between depressive mood and sleep inertia is still not clearly understood, and particularly, sleep inertia requires the interest of pharmaceutical firms to be effectively treated. The role of increased fatigue is not only specific for central disorders of hypersomnolence, but also affects other diseases. Our study illustrated that more attention should be focused on pathophysiological mechanisms and associations of these factors, particularly in women, in whom fatigue, depression, and sleep inertia prevail.

## Figures and Tables

**Figure 1 brainsci-12-01491-f001:**
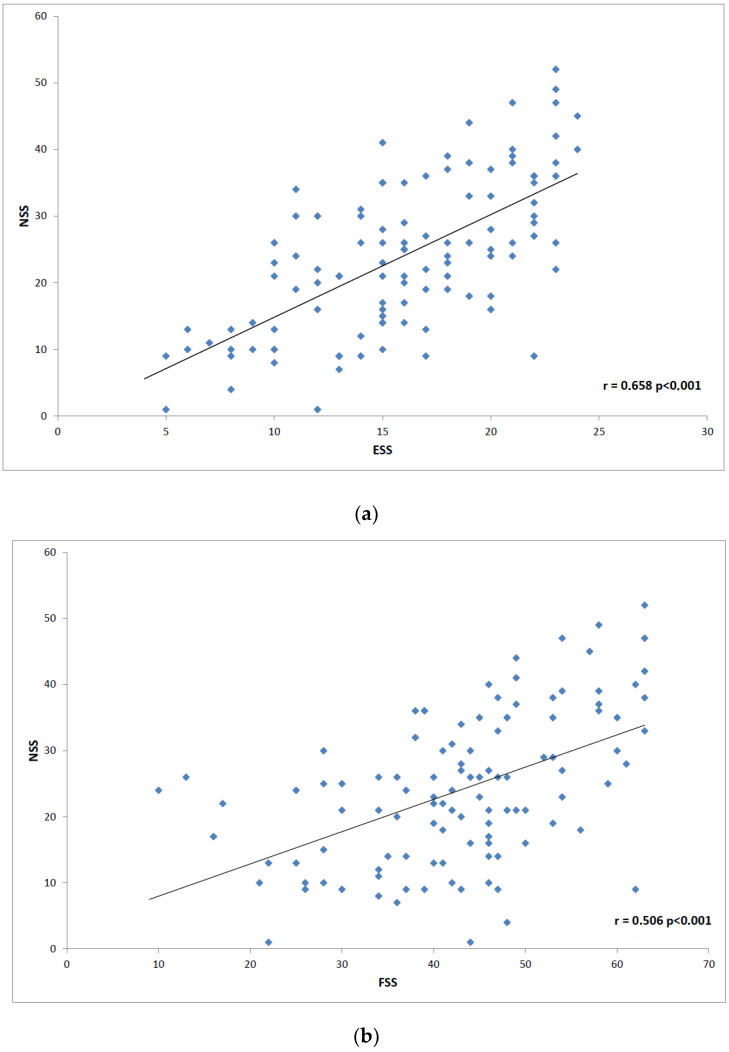
Scatter plot graphs: (**a**) shows correlation between the Narcolepsy Severity Scale (NSS) and Epworth Sleepiness Scale (ESS); (**b**) illustrates correlation between NSS and Fatigue Severity Scale (FSS).

**Figure 2 brainsci-12-01491-f002:**
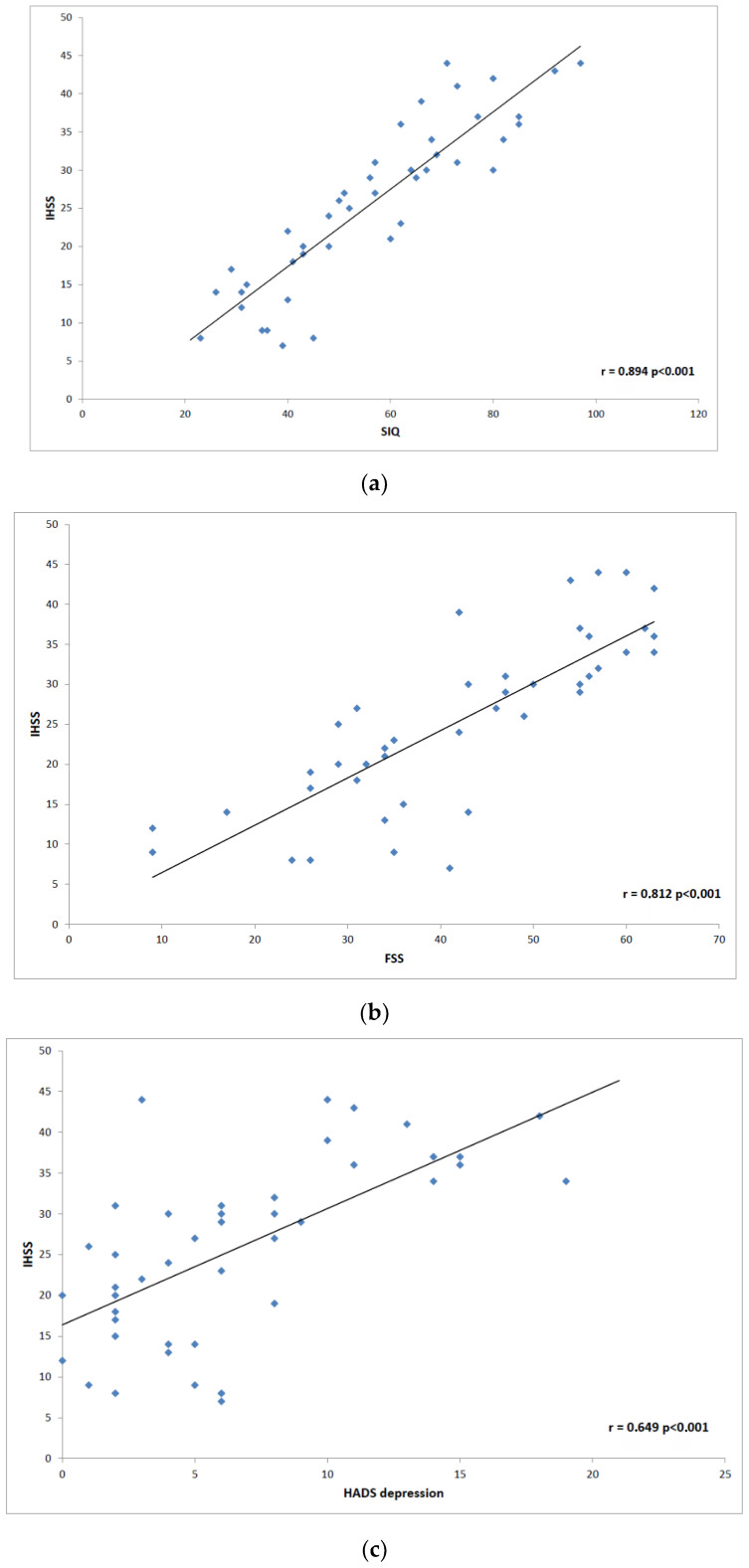
Scatter plot graphs: (**a**) shows correlation between the Idiopathic Hypersomnia Severity Scale (IHSS) and Sleep Inertia Questionnaire (SIQ); (**b**) correlation between IHSS and Fatigue Severity Scale (FSS); (**c**) correlation between IHSS and Hospital Anxiety Depression Scale (HADS) depression.

**Table 1 brainsci-12-01491-t001:** Demographic data and questionnaire results.

Parameters	NT1	NT2	IH	Total	Sign.
Age (years)	33 ± 12.7	35.5 ± 14.3	41 ± 13.2	35.45 ± 13.39	NT1 vs. NT2 NS NT1 vs. IH ** NT2 vs. IH NS
N women (%)	87 53 (61%)	22 13 (59%)	39 27 (69%)	148 93 (63%)	NS
BMI	30.3 ± 7.6	25.7 ± 4.8	26.5 ± 6.0	28.64 ± 7.16	NT1 vs. NT2 ** NT1 vs. IH ** NT2 vs. IH NS
HADS Anxiety	5.78 ± 3.90	6.91 ± 4.23	7.15 ± 4.52	6.32 ± 4.14	NT1 vs. NT2 NS NT1 vs. IH NS NT2 vs. IH NS
HADS Depression	4.61 ± 3.59	6.45 ± 4.96	6.69 ± 5.18	5.45 ± 4.36	NT1 vs. NT2 NS NT1 vs. IH * NT2 vs. IH NS
ESS	16.83 ± 4.68	12.95 ± 4.37	13.74 ± 4.78	15.44 ± 4.93	NT1 vs. NT2 ** NT1 vs. IH ** NT2 vs. IH NS
NSS	26.47 ± 10.87	14.91 ± 8.17	-	23.91 ± 11.41	NT1 vs. NT2 ***
IHSS	-	-	25.93 ± 10.96	25.93 ± 10.96	-
FSS	43.57 ± 11.46	41.09 ± 12.91	42.0 ± 15.84	42.79 ± 12.90	NS
SIQ	47.79 ± 14.86	53.24 ± 19.33	57.56 ± 19.72	51.16 ± 17.34	NT1 vs. NT2 NS NT1 vs. IH ** NT2 vs. IH NS

Mean ± SD Sign.: * *p* < 0.05 ** *p* < 0.01 *** *p* < 0.001. NT1 = Narcolepsy type 1, NT2 = Narcolepsy type 2, IH = Idiopathic hypersomnia, BMI = Body Mass Index, HADS = Hospital Anxiety and Depression Scale, ESS = Epworth Sleepiness Scale, NSS = Narcolepsy Severity Score, IHSS = Idiopathic Hypersomnia Severity Scale, FSS = Fatigue Severity Scale, SIQ = Sleep Inertia Questionnaire.

**Table 2 brainsci-12-01491-t002:** Questionnaire results in idiopathic hypersomnia without (IH1) or with (IH2) long sleep duration/24 h.

	IH1 N = 21	IH2 N = 18	Sign.
IHSS	21.71 ± 9.64	33.71 ± 7.77	*p* < 0.001
HADS Anxiety Mean ± SD	6.43 ± 3.93	8.00 ± 5.11	NS
HADS Depression Mean ± SD	5.62 ± 4.93	7.94 ± 5.31	NS
FSS Mean ± SD	36.67 ± 14.96	48.22 ± 14.89	*p* < 0.05
ESS Mean ± SD	12.29 ± 4.53	15.44 ± 4.62	*p* < 0.05
SIQ Mean ± SD	50.8 ± 20.6	65.5 ± 15.7	*p* < 0.05

NT1 = Narcolepsy type 1, NT2 = Narcolepsy type 2, IH = Idiopathic hypersomnia, HADS = Hospital Anxiety and Depression Scale, ESS = Epworth Sleepiness Scale, NSS = Narcolepsy Severity Score, IHSS = Idiopathic Hypersomnia Severity Scale, FSS = Fatigue Severity Scale, SIQ = Sleep Inertia Questionnaire.

**Table 3 brainsci-12-01491-t003:** Sex differences between questionnaire data in different diagnostic groups.

	NT1 N = 87			NT2 N = 22			IH N = 39		
	W N = 53	M N = 34	Sign.	W N = 13	M N = 9	Sign.	W N = 27	M N = 12	Sign.
HADS Anxiety Mean ± SE	6.15 ± 0.56	5.18 ± 0.71	NS	8.46 ± 1.13	4.67 ± 1.35	*p* < 0.05	7.74 ± 0.78	5.83 ± 1.17	NS
HADS Depression Mean ± SD	4.85 ± 0.59	4.24 ± 0.74	NS	7.69 ± 1.18	4.67 ± 1.42	NS	7.30 ± 0.82	5.33 ± 1.23	NS
FSS Mean ± SD	44.30 ± 1.72	42.44 ± 2.15	NS	48.54 ± 3.47	30.33 ± 4.18	*p* < 0.01	43.48 ± 2.41	38.67 ± 3.62	NS
ESS Mean ± SD	17.40 ± 0.63	15.94 ± 0.78	NS	15.08 ± 1.27	9.89 ± 1.52	*p* < 0.01	13.89 ± 0.88	13.42 ± 1.32	NS
SIQ Mean ± SD	48.85 ± 2.30	46.15 ± 2.87	NS	60.42 ± 4.82	43.67 ± 5.57	*p* < 0.05	59.30 ± 3.22	53.67 ± 4.82	NS

NT1 = Narcolepsy type 1, NT2 = Narcolepsy type 2, IH = Idiopathic hypersomnia, W = women, M = men, HADS = Hospital Anxiety and Depression Scale, ESS = Epworth Sleepiness Scale, NSS = Narcolepsy Severity Score, IHSS = Idiopathic Hypersomnia Severity Scale, FSS = Fatigue Severity Scale, SIQ = Sleep Inertia Questionnaire.

**Table 4 brainsci-12-01491-t004:** Psychiatric comorbidities in different diagnostic groups.

Psychiatric Disease	NT1 N = 87	NT2 N = 22	IH N = 39	Total N = 148
Mixed anxiety-depressive disorder	19 (21.9%)	8 (36.4%)	16 (41.0%)	43 (29.0%)
Depression	3 (3.5%)	0 (0.0%)	3 (7.7%)	6 (4.1%)
General anxiety syndrome	5 (5.8%)	0 (0.0%)	2 (5.1%)	7 (4.7%)
Attention deficit hyperactivity disorder	2 (2.3%)	2 (9.1%)	0 (0.0%)	4 (2.7%)
Pervasive developmental disorders	2 (2.3%)	0 (0.0%)	0 (0.0%)	2 (1.3%)
Schizophrenia	2 (2.3%)	0 (0.0%)	0 (0.0%)	2 (1.3 %)
Total	33 (38.1 %)	10 (45.5%)	21 (53.8%)	64 (43.1%)

Note: Two patients had a dual diagnosis.

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
