# Peer review of "Central Disorders of Hypersomnolence: Association with Fatigue, Depression and Sleep Inertia Prevailing in Women"

_brainsci, 2022, doi:10.3390/brainsci12111491_

Round 1

Reviewer 1 Report

The manuscript reports an interesting study about the associations between hypersomnolence and other symptoms crucial in the everyday life. The results are interesting and well-presented. I have only a few comments for the authors to improve their manuscript and to clarify some points that should be improved. 

-  please avoid using "follow" because it might confuse readers (lines 95 and 105). Indeed, your paper reports a cross-sectional study (even if some evaluation needed a longer record time), and "follow" might be interpreted as a longitudinal evaluation.

- several demographic variables significantly differ between groups (age, BMI). Have you evaluated the results including them as covariates? 

- please, be more detailed about the enrollment of the participants. Are they patients that asked for clinical purposes? In that case, the authors should include in their manuscript the limit of self-selection of participants. This aspect might reduce the role of differences in demographic variables and might become more necessary to use them as covariates.

- why did only one group fill out the IHSS questionnaire? It is quite useless now.

- Are there any differences between IH1 and IH2 groups as regards age, BMI, and sex?

- samples sizes are different between groups, I think you should use only a non-parametric approach for your statistical analyses

- I have specific concerns about point 3.3. Reading the text, it seems that you evaluated psychiatric conditions after evaluating the presence of high levels of depression or anxiety. In that case, were participants informed? Because it might be an ethical problem. Otherwise, you should change the text. Moreover, have you evaluated if the results are stable even without these participants (I think it is not possible for anxiety but it is possible for schizophrenia and developmental disorders)?

- Have you evaluated if participants were under medications? Was it stable?

- Please include a statement of the limits of your manuscript. 

- Lines 280-284: I think that without any information about medications in participants, it is impossible to be so confident about the results. Moreover, I think the self-report nature of the questionnaire might induce the authors to step down their conclusions more interlocutory way.

Author Response

Many thanks for your remarks and comments

Reviewer 2 Report

This is an interesting study evaluating how fatigue, depression and sleep inertia influence disease severity. Patients were grouped into three main groups: narcolepsy type 1, narcolepsy type 2, and idipathic hypersomnia. The paper is well written and of interest for the readers. However, several minor changes should be made.

Abstract

1- The main description of the study design is lacking in the methods of the abstract section.

2-The authors reported "HADS anxiety and depression scales" were higher... I prefer to say "HADS anxiety and depression scores were higher.

Introduction

1- Why are the authors focusing the investigation on these three main hypersomnolence groups? This should be described before the aims of the study.

2- The main aim of the study was to follow a group of patients or to study the influence of several variables on disease severity? Please, clarify the primary and secondary objectives.

Material and methods

1- I recommend to rename the 2.1. subsection. It should be participants and study design.

2- 2.2. should be better renamed to Diagnostic Interview and Assessment Instruments.

3- I recommend to introduce the scales and questionnaires that the authors used. Why did they use the Epworth Sleepiness Scale? Why not the Pittsburg Insomnia Rating Scale?

4- For the statistical analyses, which program did the authors use?

Results

1- subsection 3.2. should be renamed. Are the authors reporting gender differences in assessment scales?

2- 3.3. subsection should be also renamed. It is better to say "Comorbid psychiatric diseases".

Discussion

1- A limitations and strengths section is needed at the end of the discussion.

Author Response

(The authors gave the same response as above.)

Round 2

Reviewer 1 Report

I think the authors have addressed all my concerns.